# Mitochondrial Genotoxicity of Hepatitis C Treatment among People Who Inject Drugs

**DOI:** 10.3390/jcm10214824

**Published:** 2021-10-20

**Authors:** Mélusine Durand, Nicolas Nagot, Quynh Bach Thi Nhu, Roselyne Vallo, Linh Le Thi Thuy, Huong Thi Duong, Binh Nguyen Thanh, Delphine Rapoud, Catherine Quillet, Hong Thi Tran, Laurent Michel, Thanh Nham Thi Tuyet, Oanh Khuat Thi Hai, Vinh Vu Hai, Jonathan Feelemyer, Philippe Vande Perre, Don Des Jarlais, Khue Pham Minh, Didier Laureillard, Jean-Pierre Molès

**Affiliations:** 1Pathogenesis and Control of Chronic and Emerging Infections, University of Montpellier, INSERM, EFS, University of Antilles, 34394 Montpellier, France; n-nagot@chu-montpellier.fr (N.N.); roselyne.vallo@umontpellier.fr (R.V.); delphine.rapoud@inserm.fr (D.R.); catherine.quillet@inserm.fr (C.Q.); p-van_de_perre@chu-montpellier.fr (P.V.P.); didier.laureillard@chu-nimes.fr (D.L.); jean-pierre.moles@inserm.fr (J.-P.M.); 2Faculty of Public Health, Hai Phong University of Medicine and Pharmacy, Haiphong 04212, Vietnam; btnquynh@hpmu.edu.vn (Q.B.T.N.); lethuylinh1189@gmail.com (L.L.T.T.); dthuong@hpmu.edu.vn (H.T.D.); nthanhbinh@hpmu.edu.vn (B.N.T.); tranthihong42@gmail.com (H.T.T.); pmkhue@hpmu.edu.vn (K.P.M.); 3CESP UMR1018, Paris Saclay, Pierre Nicole Center, French Red Cross, 75005 Paris, France; laurent.michel@croix-rouge.fr; 4Supporting Community Development Initiatives, Hanoi 11513, Vietnam; thanhnham@scdi.org.vn (T.N.T.T.); oanhkhuat@scdi.org.vn (O.K.T.H.); 5Infectious & Tropical Diseases Department, Viet Tiep Hospital, Haiphong 04708, Vietnam; vinhvuhai@gmail.com; 6School of Global Public Health, New York University, New York, NY 10003, USA; jf3880@nyu.edu (J.F.); don.desjarlais@nyu.edu (D.D.J.); 7Infectious & Tropical Diseases Department, Caremeau University Hospital, 30029 Nîmes, France

**Keywords:** mitochondria, genotoxicity, HCV treatment, drug users

## Abstract

Antiviral nucleoside analogues (ANA) are newly used therapeutics acting against the hepatitis C virus (HCV). This class of drug is well known to exhibit toxicity on mitochondrial DNA (mtDNA). People who inject drugs (PWID) are particularly affected by HCV infection and cumulated mitotoxic drug exposure from HIV treatments (antiretrovirals, ARV) and other illicit drugs. This study aims to explore the impact of direct-acting antiviral (DAA) treatments on mtDNA among PWID. A total of 470 actively injecting heroin users were included. We used quantitative PCR on whole blood to determine the mitochondrial copy number per cell (MCN) and the proportion of mitochondrial DNA deletion (MDD). These parameters were assessed before and after DAA treatment. MDD was significantly increased after HCV treatment, while MCN did not differ. MDD was even greater when subjects were cotreated with ARV. In multivariate analysis, we identified that poly-exposure to DAA and daily heroin injection or regular consumption of methamphetamines were positively associated with high MCN loss while DAA and ARV treatments or methadone use were identified as risk factors for having mtDNA deletion. These observations deserve attention since they were previously associated with premature cell ageing or cell transformation and therefore call for a long-term follow-up.

## 1. Introduction

For a decade, direct-acting antivirals (DAA) have been able to cure most HCV-infected patients with a strong efficacy against all HCV genotypes and a strong safety profile [1]. This breakthrough has allowed the World Health Organisation (WHO) to set the goal of HCV elimination by 2030 (90% reduction in HCV incidence, 65% reduction in mortality, 80% reduction in HCV-positive people treated) [2].

All DAA regimens include inhibitors of NS3/4A protease and NS5A or NS5B polymerase. NS5B inhibitors can be antiviral nucleoside analogues (ANA) or non-nucleotide inhibitors. Numerous candidates have failed to complete the drug development process and have had to be stopped during phase II or III of clinical trials because of elevated toxicity, as is the case for balapiravir, for example, which caused serious adverse hematological, infectious, and ocular events, or INX-08189 (BMS-986094), which appeared cardiotoxic [3,4,5]. These molecules, which acted as obligate or non-obligate chain terminators during viral replication, later appeared to also be substrates of the mitochondrial RNA polymerase (PolRMT), responsible for the mitochondrial DNA (mtDNA) transcription but also acting as primase to initiate mtDNA replication. Such an off-target effect could explain the toxic outcomes described above [6,7]. The PolRMT, unlike nuclear polymerases, is more sensitive to genotoxicants because of the lack of editing functions [6]. Consequences are mtDNA alterations and cellular stress, which can lead to short and/or long-term impairment of cellular functions. Short-term consequences can be as serious as pancreatitis or lactic acidosis, while long-term consequences are related to premature ageing and chronic inflammation (termed inflammageing) and age-related diseases [6,8,9,10]. Sofosbuvir, one of the backbones of the HCV regimen, has been assessed for its genotoxicity; results showed a low affinity for PolRMT and poor mitochondrial toxicity but in cell-based assays or in vitro enzymatic assays, as reported in its monography (2013) [5,11]. To date, in vitro safety studies of combinations of DAA as well as the stricto sensus evaluation of the acute or long-term mitochondrial genotoxicity of the HCV DAA regimen in humans are missing. More recent studies have questioned the complete safety of DAA treatments. The first study reported that sofosbuvir had a hormetic effect on rat hepatocytes. At low concentrations (25–100 µM), this molecule induces oxidative stress as well as a decline in mitochondrial membrane potential. However, at high concentrations (400 µM), deleterious effects were reversible and rather hepatoprotective, also demonstrating antioxidant properties [12]. These observations have been further substantiated by the most recent investigations in yeast models. Sofosbuvir or daclatasvir reduced the relative mtDNA content, altered the mitochondria integrity, and impaired mitochondria respiration [13]. The liver concentration of sofosbuvir during treatment (around 70 µM) is compatible with the concentration of the abovementioned experiments [14,15].

The safety of the treatment also requires the consideration of the various characteristics of patients who will be treated. The HCV epidemic is concentrated in certain populations such as people who inject drugs (PWID), estimated to be more than 15.6 million worldwide [16]. More than half of them have been exposed to HCV, thereby making this population a non-negligible reservoir in terms of public health and HCV elimination. Moreover, HIV/HCV coinfections are very high among PWID living with HIV (PLHIV), from 72% to 91% [17,18]. This population is particularly exposed to several potent mitotoxic drugs, including those used in HIV treatments, which have been implicated in severe side effects such as myopathies, peripheral neuropathies, pancytopenia, or lipodystrophy, with mitochondrial dysfunction through off-target effects on mitochondrial polymerases as the underlying mechanism [19]. Furthermore, drug consumption is another source of mtDNA impairment: (i) heroin injectors showed mtDNA deletions and a decrease in mtDNA copy number per cell in blood, consistent with mtDNA damages observed in the blood and the hippocampus of rats having received heroin doses twice a day [20]; (ii) methamphetamine also causes in vitro mtDNA damages and oxidative stress on human primary neurons, human primary pulmonary endothelium, and lymphocytes T cells, and mtDNA damages have also been reported in postmortem brain samples of methamphetamine-using PLHIV [21,22,23,24]. Taken together, these elements raise the question of whether DAAs, which are clinically very well tolerated, may have a cumulative or a synergic effect on mtDNA damages among poly-exposed PWID.

In this study, we aimed to ascertain the genotoxicity of DAA treatments consisting of different combinations of sofosbuvir, daclatasvir, and ribavirin, among PWID. Our two outcomes re the evolution of the mtDNA copy number per cell (MCN) and the proportion of mtDNA deletion (MDD) in the blood of active drug users before and after DAA treatments. We then aimed to determine whether poly-exposure to various substances and medications had an impact on these mtDNA parameters, to in fine define a subset of PWID requiring a closer follow-up.

## 2. Materials and Methods

### 2.1. Study Design and Study Population

We set a before-after observational study among subjects drawn from the DRIVE-C survey, which aimed to implement a massive HCV test-and-treat approach to control the HCV epidemic among PWID in Haiphong, Vietnam. The full study protocol is published elsewhere [18]. All study participants signed an informed consent form that authorises the use of the banked samples for sub-studies related to the HIV and HCV epidemic among PWID. Briefly, PWID were prescreened for chronic HCV. HCV-RNA-positive subjects were eligible for the study and further addressed for clinical and biological parameters. They completed questionnaires to collect information on socio-demographic and health data, drug consumption habits, sexual, or other behaviours linked with HCV infection during a face-to-face interview with a community support peer. If coinfected with HIV or HBV and not receiving treatments, they first initiated these treatments before initiating HCV treatments at a minimum of three months after. All chronic HCV patients received a 12-week treatment consisting of either sofosbuvir 400 mg and daclatasvir 60 mg (SOF400/DCV60) or sofosbuvir 400 mg and daclatasvir 90 mg (SOF400/DCV90) for those with an HIV infection treated with efavirenz or nevirapine. Ribavirin (RBV) was added to the regimen for patients with cirrhosis defined by a transient elastography (or FibroScan^®^, Echosens, France) value >12.5 kPa. Drug dispensation was adjusted to each patient, and adherence was evaluated by pill count. At the end of the treatment, PWID returned for a follow-up visit at week 12 and were administered the same questionnaire as for the prescreening step. This study complies with the Declaration of Helsinki and Good Clinical Practice, was approved by the DRIVE-C research programs Scientific Advisory Board (NCT03537196), and subsequently by the Institutional Review Board of the Haiphong University of Medicine and Pharmacy, Vietnam (#01/HPUMPRB).

### 2.2. Sample Collection and DNA Extraction

During each visit, whole blood was collected on dried blood spot cards (DBS, WhatmanTM 903, GE Healthcare Bio-Sciences Corp., Chicago, IL, USA), dried, and then stored in a sample repository at −80 °C. Whole blood DNA was extracted from one spot using a QIAamp DNA Mini Kit in a Qiacube automate (QIAGEN), following the manufacturer’s instructions. DNA extracts were then stored at −80 °C.

### 2.3. Mitochondrial Genotoxicity Assays

Mitochondrial copy number (MCN) was assessed by real-time quantitative PCR using QuickScanTM Mitox Kit (Primagen, Amsterdam, The Netherlands), following the manufacturer’s instructions. In short, 2.5 µL of sample extracts were amplified with primers targeting the mtDNA rRNA 16S and the genomic DNA (gDNA) snRNP U1A gene and LightCycler^®^ 480 SYBR Green I Master Mix (Roche, Germany). PCRs were performed with a LightCycler^®^ 480 Instrument (Roche, Germany) with the following program: five minutes at 95 °C, 45 cycles of 95 °C for 10 s, 60 °C for 20 s and 72 °C for 20 s. In each plate, standard curves were obtained with calibrators, ranging from 625 to 10,000 copies per well for gDNA and from 7.5 × 10^4^ to 2.5 × 10^6^ copies per well for mtDNA. All plates contained a no-template control, one internal positive control from one healthy individual, and DNA samples in monoplicate. For each sample, the MCN was calculated from the Ct values and the standard curves after considering that each cell has two copies of gDNA and expressed in copies/cell (c/cell). A plate was considered valid if the Ct of the internal DNA control was within the mean ± 2SD previously determined during repeatability and reproducibility assays (data not shown). Sample results were considered valid if their Ct were under 30.

mtDNA deletion (MDD) was assessed using primers and probes previously described by Belmonte et al. [25]. Duplex real-time Taqman^®^ probe quantitative PCRs targeted one constant region of the mtDNA—namely, the ND1 gene and one region which encompasses more than 85% of the reported deletions so far—namely. the ND4 gene [26]. The reaction mixture consisted of 5 µL of DNA extracts, 10 µL of LC^®^ 480 probes Master Mix, and primers and probes. Forward primers and probes were added for a final concentration of 100 nM, while reverse primers were added for a final concentration of 250 nM. Each sample was assayed in duplicate. PCRs were performed with a LightCycler^®^ 480 Instrument (Roche, Germany) with the following program: five minutes at 95 °C, 45 cycles of 10 s at 95 °C, 30 s at 60 °C and 1 s at 72 °C and a cooling step for 10 s at 40 °C. Each plate contained a no-template control, one calibrator of DNA extracted from plasma-rich platelets (PrP, 10 ng per well), and one DNA control from a healthy individual extracted from DBS. Colour compensation was applied to each run between FAM and VIC fluorochromes. The mean efficacy of the PCR was 2.06 ± 0.06 for ND1 and 1.81 ± 0.14 for ND4, as assessed during the repeatability and reproducibility assays (data not shown). Plates were validated if the 2^−ΔΔCt^ of the DNA control was within the mean ± 2SD of the 2^−ΔΔCt^ of each plate. The percentage of mtDNA deletion was obtained by relative quantification with the 2^−ΔΔCt^ method using PrP as the calibrator [27]. The sample was considered valid if Ct were under 30 in both wells and if the variation of Ct between the two duplicates did not exceed 5%. In this case, the sample was repeated and next excluded from the analysis if still discordant. Flowchart on sample selection is provided in the supplementary data (Appendix A).

### 2.4. Statistical Analysis

Assessment of DAA’s mitochondrial toxicity was based on samples from 470 randomly selected subjects from the DRIVE-C study. This population size allowed us to significantly discern a 15.4% and 9.0% difference in MCN and MDD, respectively, on paired samples, with a power of 80% and an alpha risk of 5%. Power calculations were based on preliminary experiences (MCN mean (SD): 581.6 (491.7) c/cell; MDD mean (SD): 0.84 (0.076)) and assuming that MCN and MDD values do not vary significantly within the time frame of the study for uninfected PWID.

Baseline characteristics were described as frequencies with percentages for categorical variables or means with standard deviations (SD) for continuous variables. MtDNA parameters were expressed as percentage variation before and after treatment, with their 95% confidence interval (ΔMCN and ΔMDD, % (95% CI)).

For the study of risk factors associated with high loss of mtDNA copy number or high accumulation of mtDNA deletions, PWID were stratified into two classes depending on their ΔMCN and then ΔMDD results, using a threshold set at the first tertile value of the pooled data, equivalent to ≤−39% for ΔMCN and to ≤−32% for ΔMDD. We first performed univariate analysis using Pearson’s chi-squared or Fisher exact tests (alpha risk = 5%) to assess the association between the two outcomes and the following variables: ‘sex’, ‘age’, ‘number of years of heroin injection ‘, ‘frequency of heroin injection during the month preceding the interview’, ‘methamphetamine consumption during the month preceding the interview’, and ‘cigarette smoking’, which were self-declared. Additionally, ‘methadone intake’, ‘methamphetamine consumption’, ‘HIV coinfection’, and ‘HBV coinfection’ which relied on the results of urine/blood tests. ‘Hazardous drinking’ was assessed by the AUDIT-C questionnaire. A score greater or equal to four in men, or three in women, is considered to be positive for hazardous drinking and/or active alcohol use disorders [28]. ‘DAA ± ARV treatments’ were recorded from the DRIVE-C database. All variables with a *p* ≤ 0.20 were then included in multivariate models. The proportions of the two outcomes were superior to 5% in this population; thus, we performed a logarithmic binomial regression. For DAA treatments, we first considered four categories: SOF400/DCV60, SOF400/DCV90, SOF400/DCV60/RBV, and SOF400/DCV90/RBV. However, treatments with DCV90, HIV coinfection, and ARV treatments were collinear, as all HIV-infected PWID received ARV treatment which subsequently justifies the increased dosage of DCV to 90 mg due to the potency of efavirenz and nevirapine as an inducer of cytochrome P450-3A4. Therefore, we sequentially only included ‘treatment with DAA and ARV’ in the multivariate analyses. The final model was constructed in a backwards stepwise manner, considering the smallest Akaike information criteria (AIC). Statistical analysis was performed using SAS^®^ studio university (Copyright © 2012–2020, SAS Institute Inc., Cary, NC, USA) and R (v3.6.0).

## 3. Results

### 3.1. Study Population and Characteristics

Among the 979 PWID who initiated treatment, 470 were randomly selected. Of those, 451 completed treatment and had an available sample at the end of treatment. MCN-paired samples and MDD-paired samples were obtained for 418 and 332 patients, respectively (Appendix A).

At baseline (Table 1), PWID were almost exclusively men (95.7%), with a mean age of 42.0 years. All participants actively injected heroin. Most PWID injected heroin on a daily basis (70.3%), had multiple drug exposures notably combined with methamphetamines (66.5%), smoked tobacco (96.6%), and had, to a much lesser extent, excessive alcohol intake (5.5%). Consumption of other drugs was limited. Finally, half of the participants reported being on methadone-assisted therapy (50.5%).

### 3.2. Impact of DAA Treatments on mtDNA Parameters

Overall, DAA treatments did not significantly modify the MCN (3.4% of variation) but increased MDD proportion by about 16% (Table 2). However, it is worth mentioning that 27.3% of the PWID showed a 50% or more reduction in MCN and that similarly, 13.9% of the PWID accumulated more than 50% of MDD (data not shown).

Furthermore, taken individually, each type of treatment showed some differences. SOF400/DCV60 reduced the MCN, while the other treatments did not. Regarding MDD, the impact of treatments containing DCV90 was higher than those with DCV60 (−25.6% and −31.8% vs. −9.8% and −10.3%). Finally, the variation in the two parameters was not correlated (Pearson correlation coefficient *r* = 0.04; *p* = 0.47).

### 3.3. Factors Associated with High MCN Loss

We conducted stratified analyses to further address the risk factors of being a ‘PWID with high MCN loss’. Having ‘over 46 years of age’, ‘daily heroin injection’, and ‘consuming methamphetamine at least four times per month’ were associated with an increased risk of mtDNA copy number loss (aRR = 1.49 (.02; 2.18); aRR = 1.42 (1.02; 1.98); aRR = 1.59 (1.09; 2.23), respectively) (Table 3). Additionally, hazardous drinking tended to increase the risk of having less MCN, though the relationship was not statistically significant (aRR = 1.55 (0.99; 2.43)).

### 3.4. Factors Associated with High Accumulation of MDD

We conducted stratified analyses to further address the risk factors of being a ‘PWID with a high accumulation of MDD’. The analysis showed that ‘being cotreated with ARV’ and ‘methadone use’ were positively associated with high accumulation of MDD in PWID (aRR = 1.41 (1.03; 1.93); aRR = 1.53 (1.01; 2.32), respectively). Other exposures were not associated with MDD (Table 4).

## 4. Discussion

Herein, we report the mitochondrial genetic damages brought forth by HCV DAA regimens among PWID. Overall, the MCN remained unchanged, whereas the proportion of detectable mtDNA deletion was increased after treatment. However, we identified that poly-exposures to DAA and daily heroin injection or DAA and regular consumption of methamphetamine were positively associated with high MCN loss and that poly-exposure to DAA and ARV, or DAA and methadone were aggravating factors for the acquisition of mtDNA deletions.

Decrease in mtDNA copy number per cell and mtDNA deletion have been proposed as proxy indicators for mtDNA dysfunctions. Both have been reported among heroin users but also for age-related and chronic diseases such as Alzheimer’s disease, diabetes, and cancers [20,29]. Chronic inflammation results in mtDNA damage acceleration, which, in turn, feeds the inflammation, partly explaining the premature ageing observed in chronic diseases. Accelerated ageing has also been reported in chronic infections such as HIV [30]. HCV itself reduces MCN through the induction of systemic inflammation [31]. Moreover, it has been reported that HCV patients with HCC had less MCN than HCV-cirrhotic patients, suggesting that MCN could be a marker for HCV disease progression as well as a marker for cancer. However, in some studies, treatments were able to slow down MCN loss [32,33]. PWID with a chronic infection analysed herein showed a MCN that was not significantly different from PWID with no HCV infection (n = 260) (median (95% CI) 486.1 cp/cell (456.3; 516.6) vs. 439.9 cp/cell (405.9; 466.8), *p* = 0.25, data not shown). While DAA treatments had no impact on MCN, our multivariate analysis showed that exposure to DAA combined with daily heroin injection or regular consumption of methamphetamines were positively associated with high MCN loss when adjusted for patient age. Heroin and MCN were already linked, as heroin users presented lower levels of MCN than non-users [20]. However, this type of association has never been demonstrated in the blood of methamphetamine users, only on cultured human cells through increased oxidative stress and associated mitochondrial damage [21,22,23].

MtDNA mutations are responsible for inherited mitochondrial diseases but are also acquired as they accumulate physiologically with age. However, this accumulation is enhanced under certain conditions such as chronic inflammation and oxidative stress. In this case, reactive oxygen species (ROS) resulting from oxidative stress can alter mtDNA, due to its close proximity and its lack of DNA repair mechanisms, which leads to an accumulation of mutations and deletions. MtDNA impairment can also be the result of the direct action of mitotoxic drugs. Many antiviral nucleoside analogues are mistakenly integrated by mitochondrial polymerases in de novo synthesised DNA or RNA due to their structural analogy with natural nucleosides. PWID with a chronic infection analysed herein showed a MDD that was not significantly different from PWID with no HCV infection (n = 260) (median (95% CI) 0.72 (0.70; 0.75) vs. 0.69 (95% CI: 0.66; 0.76), *p* = 0.25, data not shown). Our results demonstrate a clear increase in the proportion of mtDNA deletion after DAA treatments only, and even more so among HIV-coinfected patients receiving ARV treatment. Nucleoside reverse-transcriptase inhibitor (NRTI) is a well-known mitotoxic drug acting on the mitochondrial DNA polymerase γ. Even if former ARV treatments with the highest side effects have now been replaced by less toxic molecules such as lamivudine or tenofovir, their impact on mitochondrial genotoxicity still persists and is suggested to participate in premature ageing [34,35]. Sofosbuvir, an NS5B HCV polymerase inhibitor, is the only ANA available today, as all of the other ANAs developed against HCV have shown mitochondrial toxicity levels that are too elevated. Sofosbuvir showed very little affinity towards PolRMT during in vitro cell-based experiments [5], though recent studies report a hormetic effect on hepatocytes [12] and a clear mtDNA genotoxicity in the yeast model [13]. To date, mitochondrial genotoxicity has not been assessed in vitro nor in vivo for DAA in combination with another DAA, or with ARV, and even less with poly-exposure to psychoactive substances. These findings suggest that cumulative HCV and HIV treatments synergise mtDNA genotoxicity. In addition, an association between MDD acquisition and methadone treatment was observed, which, to our knowledge, has never been reported in prior studies. However, during in vitro experiments, methadone induced necrotic-like cell death along with mitochondrial damage [36] and negatively changed mitochondrial morphology [37]. This toxic effect could be the result of drug–drug interactions such as methadone and daclatasvir which are both substrates of the PgP protein [38]. A recent study reported that when sofosbuvir/daclatasvir and methadone were coadministered, the plasma concentration of daclatasvir was increased [39]. Therefore, methadone toxicity could be enhanced by DAAs and vice versa.

It is of note that our conclusions were challenged by the well-known confounders of mtDNA genotoxicity, i.e., ‘sex’, ‘HBV infection’, and ‘smoking cigarettes’. Given the absence of tobacco intervention during the trial, the before–after observational design of the study, and the paired analyses, these variables remained unchanged at the individual level.

This study has limitations. First, we measured the mitochondrial outcomes in whole blood, but mtDNA damages may be different in other organs, particularly those with slower cell renewal and those with a high content of mitochondria. Indeed, the accumulation of mtDNA damages was frequently higher in brain, muscle, or heart cells, compared with blood [40]. One could suggest that mtDNA damages would be also much higher in HCV-infected cells. Secondly, the integrity of the mtDNA molecules was only investigated through the search of one target and not in terms of point mutation. Other regions of the mtDNA may also be targeted for deletions, given that drug exposures were particular. The latter two aspects would require sequencing approaches. Among the strengths of our study, we can underline the large number of subjects included and the homogeneity of the population, allowing us to make clear conclusions on mtDNA parameters among PWID. These results call for new investigations among non-drug users.

Altogether, this is the first study determining mitochondrial toxicity of DAA among a population of drug users. Our findings strongly suggest that mitochondrial biogenesis is altered by HCV treatments, supported by the observation of accumulated deletions. Although examples of recovery of the mtDNA copy number per cell have been reported upon drug discontinuation, in particular with HIV drugs [41,42], the persistence of deleted mtDNA molecules may lead to bioenergetics defects, mito-ageing processes, and possibly cell transformation. We identified a subset of PWID patients taking DAA treatments at a higher risk of MDD acquisition, most probably linked to poly-exposure and to the synergetic effects of these exposures. While DAA treatments have been the tool expected to end the HCV epidemic [2], a subset of patients may deserve a closer and more extended follow-up to ensure their long-term health.

## Figures and Tables

**Table 1 jcm-10-04824-t001:** Baseline characteristics of HCV-positive PWID in Hai Phong.

	Patients with Chronic HCV Included in the Analysis (*n* = 418)
**Socio-demographic data**	
Sex, *n* (%), Male/Transgender	400 (95.7)
Age, *mean* (SD), Years	42.0 (7.5)
**Viral infections**, *n* (%)	
HIV	180 (43.1)
HBV	22 (5.3)
**Drug consumption—Heroin**, *n* (%)	
Number of years of injection	
<5 years	19 (4.5)
5 to 10 years	96 (23.0)
10 to 15 years	120 (28.7)
More than 15 years	183 (43.8)
Frequency of injection during the month preceding the interview	
Less than daily	124 (29.7)
Daily	294 (70.3)
**Methadone**, *n* (%)	
Urinary test positive	293 (70.1)
Declare being on methadone-assisted therapy	211 (50.5)
**Methamphetamine**, *n* (%)	
Urinary test positive	122 (29.2)
Declare consumption	278 (66.5)
Number of years of consumption	
≤5 years	149 (35.6)
5 to 10 years	99 (23.7)
More than 10 years	30 (7.2)
Frequency of consumption during the month preceding the interview	
<4 times per month	207 (49.5)
≥4 times per month	71 (17.0)
**Other drugs**, *n* (%)	
Cannabis	36 (8.6)
Ketamine	7 (1.7)
Ecstasy	8 (1.9)
Amphetamines	6 (1.4)
Cocaine	1 (0.2)
**Tabaco**, *n* (%)*,* Cigarette smoking	404 (96.6)
**Alcohol**, Hazardous drinking (AUDIT-C), *n* (%)	23 (5.5)

**Table 2 jcm-10-04824-t002:** MtDNA parameter variation after DAA treatments ((ETT—Baseline/Baseline)*100) in %, median (95% CI).

	ΔMCN	ΔMDD
**All DAA treatments, *n* = 418**	3.4 (−9.7; 13.6)	−15.9 (−22.1; −10.3) *
Sofosbuvir/daclatasvir 60 mg, *n* = 227	−19.1 (−30.6; 6.0)	−9.8 (−19.0; −0.4) ^†^
Sofosbuvir/daclatasvir 90 mg, *n* = 155	26.5 (3.4; 73.8)	−25.6 (−32.4; −17.4) ^§^
Sofosbuvir/daclatasvir 60 mg + ribavirin, *n* = 25	0.2 (−41.6; 60.0)	−10.3 (−17.9; 13.1) ‖
Sofosbuvir/daclatasvir 90 mg + ribavirin, *n* = 11	25.5 (−45.9; 1700.2)	−31.8 (−60.2; −0.56) ¶

ETT: end of treatment; 95% CI: distribution-free 95% confidence interval for medians; *: 86 missing values; ^†^: 51 missing values; ^§^: 27 missing values; ‖: 6 missing values; ¶: 2 missing values.

**Table 3 jcm-10-04824-t003:** Factors associated with high MCN loss.

		UNIVARIATE		MULTIVARIATE	
	*n* (%)	RR (IC95%)	*p*-Value	aRR (IC95%)	*p*-Value
**Socio-demographic data:**					
Male sex	400 (95.7)	1.00 (0.51; 1.94)	0.99	-	
Age			0.07		0.05
Less than 37 years old	95 (22.7)	Ref.		Ref.	
37 to 40 years old	77 (18.4)	1.16 (0.76; 1.77)		1.16 (0.76; 1.76)	
41 to 46 years old	127 (30.4)	0.97 (0.62; 1.53)		0.95 (0.61; 1.48)	
Over 46 years old	119 (28.5)	1.48 (1.01; 2.18)		1.49 (1.02; 2.18)	
HIV coinfection	180 (43.1)	0.81 (0.61; 1.08)	0.15	-	
HBV coinfection	22 (5.3)	1.10 (0.62; 1.94)	0.75	-	
**Treatments:**					
DAA treatments			0.08		
DAA only	252 (60.3)	Ref.			
DAA and ARV	166 (39.7)	0.77 (0.58; 1.04)		-	
Methadone *	293 (70.1)	0.81 (0.61; 1.07)	0.15	-	
**Drug/substance consumption:**					
Heroin, number of years of injection			0.78		
Less than 5 years	9 (4.5)	Ref.			
Between 5 and 10 years	96 (23.0)	0.84 (0.46; 1.52)		-	
Between 10 and 15 years	120 (28.7)	0.77 (0.43; 1.39)		-	
More than 15 years	183 (43.8)	0.75 (0.43; 1.33)		-	
Heroin, frequency of injection during the month preceding the interview			0.02		0.03
Less than daily	124 (29.7)	Ref.		Ref.	
Daily	294 (70.3)	1.47 (1.05; 2.06)		1.42 (1.02; 1.98)	
Methamphetamine consumption ^$^	122 (29.2)	1.24 (0.94; 1.64)	0.14	-	
Methamphetamine, consumption during the month preceding the interview			0.08		0.05
No consumption	140 (33.5)	Ref.		Ref.	
<4 times per month	207 (49.5)	1.26 (0.91; 1.76)		1.34 (0.97; 1.86)	
≥4 times per month	71 (17.0)	1.56 (1.06; 2.29)		1.59 (1.09; 2.33)	
Cigarette smoking	404 (96.6)	1.57 (0.57; 4.32)	0.32	-	
Hazardous drinking ^£^	23 (5.5)	1.48 (0.94; 2.31)	0.14	1.55 (0.99; 2.43)	0.08

* Methadone status determined by urine test; **^$^** methamphetamine consumption assessed by urine testing; **^£^** Hazardous alcohol consumption defined by AUDIT-C score > 4 for men or > 3 for women*;* RR: relative risk; aRR: adjusted relative risk; 95% CI: 95% confidence interval.

**Table 4 jcm-10-04824-t004:** Factors associated with high accumulation of mtDNA deletions.

		UNIVARIATE		MULTIVARIATE	
	*n* (%)	RR (IC95%)	*p*-Value	aRR (IC95%)	*p*-Value
**Socio-demographic data:**					
Male sex	323 (97.3)	1.49 (0.43; 5.11)	0.48	-	
Age			0.94		
Less than 37 years old	96 (28.9)	Ref.			
37 to 40 years old	58 (17.5)	0.90 (0.57; 1.42)		-	
41 to 46 years old	89 (26.8)	1.02 (0.66; 1.56)		-	
Over 46 years old	89 (26.8)	0.92 (0.59; 1.43)		-	
HIV coinfection	147 (44.3)	1.54 (1.13; 2.10)	0.006	-	
HBV coinfection	20 (6.0)	1.24 (0.70; 2.16)	0.49	-	
**Treatments:**					
DAA treatments			0.004		0.03
DAA	195 (58.7)	Ref.		Ref.	
DAA and ARV	137 (41.3)	1.56 (1.15; 2.12)		1.41 (1.03; 1.93)	
Methadone *	235 (70.8)	1.73 (1.14; 2.61)	0.004	1.53 (1.01; 2.32)	0.03
**Drug/substance consumption:**					
Heroin, number of years of injection			0.72		
Less than 5 years	14 (4.2)	Ref.			
Between 5 and 10 years	68 (20.5)	1.18. (0.48; 2.89)		-	
Between 10 and 15 years	88 (26.5)	0.99 (0.41; 2.43)		-	
More than 15 years	162 (48.8)	1.23 (0.52; 2.89)		-	
Heroin, frequency of injection during the month preceding the interview			0.01		0.09
Less than daily	95 (28.6)	Ref.		Ref.	
Daily	237 (71.4)	0.66 (0.49; 0.90)		0.76 (0.56; 1.04)	
Methamphetamine consumption ^$^	92 (27.7)	0.90 (0.63; 1.29)	0.56	-	
Methamphetamine, consumption during the month preceding the interview			0.12		0.08
No consumption	113 (34.0)	Ref.		Ref.	
<4 times per month	165 (49.7)	1.16 (0.83; 1.62)		1.32 (0.95; 1.84)	
≥4 times per month	54 (16.3)	0.70 (0.40; 1.23)		0.84 (0.48; 1.48)	
Cigarette smoking	321 (96.7)	1.21 (0.45; 3.22)	0.68	-	
Hazardous drinking ^£^	15 (4.5)	0.80 (0.34; 1.89)	0.60	-	
**Mitochondrial toxicity:**					
PWID with decreased MCN ^§^	89 (26.8)	1.04 (0.74; 1.46)	0.84	-	

* Methadone status determined by urine test; **^$^** methamphetamine consumption assessed by urine testing; **^£^** Hazardous alcohol consumption defined by AUDIT-C score > 4 for men or > 3 for women; ^§^ Ref. ΔMCN > −39%; RR: relative risk; aRR: adjusted relative risk; 95% CI: 95% confidence interval.

## Data Availability

The data presented in this study are available on request from the corresponding author.

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
