# Peer review of "Mitochondrial Genotoxicity of Hepatitis C Treatment among People Who Inject Drugs"

_jcm, 2021, doi:10.3390/jcm10214824_

Round 1
Reviewer 1 Report
In May 2016, the World Health Assembly adopted the first Global Health Sector Strategy on Viral Hepatitis 2016-2021. The strategy aims to eliminate viral hepatitis as a public health problem by reducing new viral hepatitis infections by 90% and reducing viral hepatitis deaths by 65% ​​by 2030.
The hope that this goal will be achieved is associated with the use of highly effective direct-acting antiviral drugs (DAA), and, in particular, nucleoside analogues. Nucleoside analogs act as chain terminators, inhibiting viral replication. However, these drugs also have side effects. Common symptoms of nucleoside analog toxicity include peripheral neuropathy, myopathy, bone marrow suppression, and pancreatitis. There is growing evidence that these analogs also act as substrates for human mitochondrial DNA polymerase, leading to inhibition of mitochondrial replication. Unlike nuclear polymerases, mitochondrial DNA polymerase is more sensitive to genotoxicants due to the lack of editing functions. The toxicity of nucleoside analogs is especially dangerous for long-term treatment of viral infections. Many of the side effects resulting from treatment with nucleoside analogs mimic the symptoms of mitochondrial diseases caused by genetic defects.
The authors aimed to establish a link between the treatment of hepatitis C with various combinations of sofosbuvir, daclatasvir, ribavirin and the mitochondrial copy number per cell (MCN) and the proportion of mitochondrial DNA deletion (MDD) among people who inject drugs (PWID) most susceptible to HCV infection. The study included the 979 PWID who initiated their treatment, 470 were randomly selected. Of those, 451 completed the treatment and had an available sample at the end of treatment. It was shown that overall MCN remained unchanged, while the proportion of detectable mtDNA deletions increased after treatment. Repeated exposure to DAA and daily injections of heroin or DAA and regular use of methamphetamine were positively associated with high loss of MCN. Repeated exposure to DAA and ARV, or DAA and methadone, were aggravating factors for the acquisition of mtDNA deletions.
This is the first study to determine the mitochondrial toxicity of DAAs among drug users.
The authors conclude that the persistence of mtDNA molecules with deletions can lead to bioenergetic defects, mito-aging processes, and, possibly, to cell transformation. PWID patients taking DAAs are at a higher risk of acquiring MDD, which is most likely due to repeated exposure and synergistic effects of these treatments. Some patients may deserve closer and longer follow-up to ensure their long-term health.
The data obtained by the authors are new and are of interest to virologists and hepatologists. The research was carried out using adequate modern methods.
Author Response
Thank you for these comments. They encourage us to move forward on this issue.
Reviewer 2 Report
The authors describe the results they found in analysing mitochondrial genotoxicity of hepatitis C treatment among PWID using the DRIVE_C survey. The topic is interesting and warrants investigations, using dry blood samples, DNA was analysed for mitochondrial copy number per cell and mitochondrial DNA deletion. The finding they have described are limited due to the criteria of the survey (ie all patients are heroin users and all patients were treated with DAAs) and maybe the time of samples, even so some conclusions could be made with a weak association of mitochondrial genotoxicity to DAA treatments.
Specific comments:
It would be useful to see true values apart from the RR values in table 3, and is anything statistically signifcant? (p-value), the CI all seems very larger to make any strong conclusions.
A control arm of patients who do not use heroin/methodone or /and do not receive DAAs would strengthen the conclusions.
Author Response
The authors describe the results they found in analysing mitochondrial genotoxicity of hepatitis C treatment among PWID using the DRIVE_C survey. The topic is interesting and warrants investigations, using dry blood samples, DNA was analysed for mitochondrial copy number per cell and mitochondrial DNA deletion. The finding they have described are limited due to the criteria of the survey (ie all patients are heroin users)
We agree with the reviewer that these findings apply only to injecting drug users and we never extrapolated these findings to non-drug users. However, given the context of HCV elimination, injecting drug users represent a non-negligible reservoir of HCV. This point is now emphasized in the introduction line 91-92 as follow: “, thereby making this population a non-negligible reservoir in terms of public health and HCV elimination”.
and all patients were treated with DAAs)
DAAs is becoming the first line of HCV treatment. Genotoxicity studies evaluating other HCV treatment such as those based on interferon may also be of interest, however they would not have the same rationale i.e. antiviral nucleoside analogues.
and maybe the time of samples,
Our objective was to evaluate the acute toxicity of the treatments. The time point “end of treatment” seemed adequate for us. Any post-treatment time points will answer a different question i.e. the persistence of the side effects. We are currently collecting samples to conduct long-term post-treatment analyses.
even so some conclusions could be made with a weak association of mitochondrial genotoxicity to DAA treatments.
Given the before-after design of the analyses, we gave a description of the impact of DAA treatment on mitochondrial genotoxicity. The mitochondrial parameters, MCN and MDD, are not expected to change for an individual over the 12-week course of the study. We added in the 2.4 subsection Statistical analysis a clarification regarding the outcome measurements, line 220-222: “and assuming that MCN and MDD values do not vary significantly within the time frame of the study for uninfected PWID”. We also added p-values in the tables for RR and aRR.
Specific comments:
It would be useful to see true values apart from the RR values in table 3, and is anything statistically signifcant? (p-value), the CI all seems very larger to make any strong conclusions.
The actual number of outcome with percentages have been added for each variable in Table 3. To avoid a too busy table, we split Table 3 into two tables with the various data requested by the reviewer. Significant p values are in bold.
A control arm of patients who do not use heroin/methadone or /and do not receive DAAs would strengthen the conclusions.
We first hypothesized that given the poly-exposition to mitotoxic drugs, the PWID population was at a higher risk of mitochondrial genotoxicity than the non-drug user population. By itself, we think that the findings herein are worth disseminating to the research community. Indeed, these findings call for new investigations as suggested by the reviewer including the impact of DAAs among non-drug users. It is now stated in the limitation section line 650 as follows: “These results call for new investigations among non-drug users.” We do not have samples at a 12-week interval for HCV-infected PWID who did not receive DAA treatments. However, PWID with no HIV or HCV infection showed a MCN and a MDD that was not significantly different from the baseline values of the HCV-infected PWID. Because the design of the study was a before-after analysis, we could not include this data in the results section. We have now specified the following in the discussion section, line 459-461: “PWID with a chronic infection analyzed herein showed a MCN that was not significantly different from PWID with no HCV infection (n=260) (median [95%CI] 486.1 cp/cell [456.3 ; 516.6] vs 439.9 cp/cell [405.9 ; 466.8], p = 0.25, data not shown).” And line 477-478: “PWID with a chronic infection analyzed herein showed a MDD that was not significantly different from PWID with no HCV infection (n=260) (median [95%CI] 0.72 [0.70; 0.75] vs 0.69 [95%CI: 0.66; median [95%CI] 0.72 [0.70; 0.75] vs 0.69 [95%CI: 0.66; 0.76], p = 0.25 0.76], p = 0.25, data not shown).”
The manuscript was edited by a native-English speaker.
Round 2
Reviewer 2 Report
The authors have addressed correctly all the concerns from the previous review.
minor error:
There is an extra bracket in Table 3 "DAA only section"
Author Response
Thank you for helping us to improve the manuscript.
The point was corrected.